# Maternal and Paternal Authoritarian Parenting and Adolescents’ Impostor Feelings: The Mediating Role of Parental Psychological Control and the Moderating Role of Child’s Gender

**DOI:** 10.3390/children10020308

**Published:** 2023-02-06

**Authors:** Yosi Yaffe

**Affiliations:** Tel-Hai Academic College, Department of Education, Upper Galilee, Qiryat Shemona 12208, Israel; yaffeyos@telhai.ac.il

**Keywords:** impostor phenomenon, parenting styles, psychological control, adolescent

## Abstract

Introduction: Recent systematic reviews about the impostor phenomenon unveil a severe shortage of research data on adolescents. The present study aimed at reducing this gap in the literature by investigating the association between maternal and paternal authoritarian parenting and impostor feelings among adolescents, while testing the mediating role played by parental psychological control and the moderating role of the child’s gender in this context. Methods: Three hundred and eight adolescents took part in an online survey, in which they reported anonymously on their impostor feelings and their parents’ parenting styles via several valid psychological questionnaires. The sample consisted of 143 boys and 165 girls, whose age ranged from 12 to 17 (*M* = 14.67, SD = 1.64). Results: Of the sample’s participants, over 35% reported frequent to intense impostor feelings, with girls scoring significantly higher than boys on this scale. In general, the maternal and paternal parenting variables explained 15.2% and 13.3% (respectively) of the variance in the adolescents’ impostor scores. Parental psychological control fully mediated (for fathers) and partially mediated (for mothers) the association between parental authoritarian parenting and the adolescents’ impostor feelings. The child’s gender moderated solely the maternal direct effect of authoritarian parenting on impostor feelings (this association was significant for boys alone), but not the mediating effect via psychological control. Conclusions: The current study introduces a specific explanation for the possible mechanism describing the early emergence of impostor feelings in adolescents based on parenting styles and behaviors.

## 1. Introduction

### 1.1. Impostor Phenomenon

The concept of impostor phenomenon refers to talented and successful individuals according to external and objective standards, who are prone to doubt their own competence as if they were a fraud who has fooled everybody else’s impression about them [1]. Rather than their personal qualities (such as intelligent and skills), people who experience impostor feelings attribute their accomplishments to external factors, such as luck, and to other factors unrelated to actual talent and ability, such as manipulation and charm [2]. The most common impostor symptoms include reluctance to accept credit for accomplishments and to internalize the sense of being talented and competent, feelings of self-doubt, a propensity to attribute success to external causes, and a chronic fear that success will not be possible to maintain [2,3]. Indeed, the impostor phenomenon construct was originally divided into three theoretical dimensions, including self-doubts about one’s own intelligence and abilities (fake), the tendency to attribute success to chance/luck (luck), and the inability to admit a good performance (discount) [4,5]. When first clinically identified and conceptualized [1], the impostor phenomenon was thought to be a gender-specific disturbance, whose origins are rooted in feminine social role stereotypes and early experiences of gender-based family dynamics. Later data, however, failed to support the phenomenon’s gender-specific premise, indicating that the impostor phenomenon is a more general problem that may similarly occur in men [2]. Indeed, the impostor phenomenon is no longer considered a gender-typical psychological issue [6]. A recent systematic review on the prevalence and predictors of the impostor phenomenon [7,8], counted 16 articles (a little less than 50% out of 33 articles that considered gender differences) that found greater symptoms of imposter feelings among women, while the majority of works (17 articles) found no gender differences. Whilst these meta-analytic data clearly indicate that the phenomenon affects both genders [7], it may imply that, subject to certain conditions, women could be somewhat more predisposed to experience impostor feelings than men. This information mostly relies on research data about adult participants, while there is a serious shortage in research studies with adolescents [9,10]. This is despite the fact that the “Impostor phenomenon feelings are already well established by adolescence and that there may be earlier ethology of impostor phenomenon” [11] (p. 402).

To date, there is growing evidence suggesting that moderate to intense impostor feelings are very prevalent phenomena in individuals of both genders [8,12,13,14,15,16], with their ratio among personnel and students varying from 9% to 82%, and on average exceeding 40%. According to Bravata and colleagues [8], the prevalence of the impostor phenomenon varies widely depending on several factors, especially the study participants (or population-based evaluation), the screening tool employed, and cut-off points used to assess symptoms. At any rate, in a view of the growing empirical information about the impostor phenomenon in recent years, it is clear now that we are facing a concerning phenomenon with potentially detrimental psychological consequences to various populations. In this regard, more and more concerned scholars are calling for the inclusion of impostor syndrome as a specific diagnostic category in the next edition of the Diagnostic and Statistical Manual (DSM) of the American Psychiatric Association [7,8].

According to the research literature, the prominent personality correlations and psychological conditions with which the impostor phenomenon often occurs as comorbid are low self-esteem, anxiety, and depression [7]. Langford and Clance (1993) [2] explained that “impostor feelings are frequently accompanied by worry, depression, and anxiety resulting from pressure to live up to one’s successful image and the fear that one will be exposed as unworthy and incompetent” (p. 495). In terms of the big five personality traits, the impostor phenomenon was found to be associated with high neuroticism and low conscientiousness, as well as with high introversion [2,5,17]. Indeed, these psychological conditions and other emotional distresses later described in the literature (such as somatic problems, emotional exhaustion, burnout, and more) may be the toll that a heightened fear over performance and intense effort spent at masking inadequacy take from those who cope with impostor feelings [18,19]. Yet, it is unclear whether the impostor phenomenon is caused by these factors, affects them, or whether they are simply co-occurring [16].

### 1.2. Parenting Styles

Parenting style is a broad construct describing stable attitudes and behaviors regarding child-rearing [20]. In her early work, Baumrind [21,22] proposed three types of parental control in child-rearing (i.e., authoritative, permissive, and authoritarian), which later became known as patterns or styles of parental authority [23]. These types of parental control generally vary by the way the parents set and enforce rules, exert power, grant autonomy, encourage verbal negotiation over decisions, use reasoning, and regulate the child’s behavior. Baumrind’s typology [23,24] also distinguishes between the three patterns or styles of parenting according to the degree and quality to which they project emotional warmth and acceptance toward the child. Later salient conceptualization of parenting styles introduced two orthogonal, independent, parental dimensions known as responsiveness and demandingness [25], from which four parenting styles were composed. These dimensions generally depict parental aspects such as warmth, affection, sensitivity, autonomy granting, reasoned communication, behavior regulation, control, confrontation, protection, and monitoring [26,27]. Responsiveness refers to the extent to which the parent shows the child love, acceptance, and affection, and giving his/her support when dealing with the child’s good and bad behaviors [26,28]. Demandingness, however, is aimed at socializing the child as part of imparting behavioral norms and maintaining the parent’s authority, which is reflected in the parent’s usage of discipline, control, and regulation of the child’s behavior [28,29]. The compositions of the responsiveness (also known as warmth) and demandingness (also known as strictness) dimensions yields four distinctive types of parenting attitudes and behaviors in child rearing (partially corresponding with Baumrind’s typology), which can reflect the family climate and the quality of parent–child relations [25,29,30]. Authoritative (i.e., warm and strict) and authoritarian (i.e., strict but not warm) parents vary on responsiveness variables such as warmth and support, whereas authoritative and indulgent (i.e., warm but not strict, similar to the labeled permissive style in the present study) parents vary on demandingness variables such as control and monitoring [24,25]. Neglectful parents are neither warm nor strict.

Research on parental socialization indicates that authoritative parenting constitutes the optimal style in affinity to psychological well-being and educational functioning of children and adolescents [27,31], especially in middleclass families from Anglo-Saxon European and north American countries. Recent research evidence from other nationalities and ethnical samples suggests, however, that the parental effect on the child’s well-being may vary by cultural contexts [30]. Indeed, the most recent evidence conducted in European and Latin American countries support the idea that the indulgent style is associated with the best child psychosocial adjustment [32,33]. Contrary to authoritative parenting, authoritarian parenting is consistently associated in the research literature with adverse psychological outcomes in children and adolescents, including externalizing and internalizing problems, and academic and psychoeducational impairments in students [31,34,35,36]. Unlike the authoritative parent, the authoritarian parent uses coercive power and assertive disciplinary practices, which include verbal hostility, arbitrary discipline, and psychological control [37]. The latter pattern was explicitly mentioned by Baumrind as parental behaviors of coercive practices, whose consequences on the child were found to be uniquely detrimental and especially predictive of internalizing problems and poor self-efficacy [38]. Parental psychological control is a form of parental control typically manifested by excessive practices of manipulation, coercion, and disrespect that intrude on the child’s psychological development [38,39]. This maladaptive pattern has been specifically associated in past research with various detrimental emotional outcomes of child development, especially anxiety and depression [40,41], and may be the concrete mechanism explaining the aversive effects of authoritarian parenting on the child’s psychological well-being reported in the literature. Indeed, some researchers have proposed that parental psychological control mediates the relationship between over-controlling parenting (such as the authoritarian style) and emotional functioning in children and adolescents [42]. Finally, a considerable body of research suggests that mothers and fathers may differ in their dominant parenting styles and practices, with the former being more supportive, responsive, and behaviorally controlling in their child-rearing orientation (i.e., authoritative parenting) and the latter being more coercive, punitive, and psychologically controlling in their child-rearing orientation (i.e., authoritarian parenting) [43]. Gender differences in parenting were also accounted for in relation to children’s and adolescents’ psychological outcomes, with somewhat more evidence suggesting possible precedence for maternal parenting over paternal parenting in both predicting externalizing and internalizing problems [44,45,46,47,48,49]. While this trend was not sufficiently consistent across studies to establish a definitive conclusion regarding the differential parental-gender effect on the child’s outcomes [35,36,50], it does warrant discrete measurements for mothers and fathers. Particularly in the context of the gender-dyadic in parent–child relations (i.e., in consideration of both parent’s and child’s gender), the findings in regard to children’s and adolescents’ psychological outcomes are considerably inconsistent across studies and require further examination in various respects [44].

### 1.3. Parenting and the Impostor Phenomenon

One of the phenomenon’s most studied etiology factors is early family relations and dynamics. The family background of people with impostor feelings has been described as unsupportive, non-expressive, conflictual, and overcontrolling [2,51]. In a recent systematic review work focusing on this very topic [10], four forms of familial and parental factors were identified as associated with the impostor phenomenon. This included parental rearing styles and behaviors, attachment styles, maladaptive parenting and parent–child relations, and familial achievement orientation. The noticeable and more promising group of studies, however, dealt with the role of parental rearing styles and behaviors in the context of the impostor phenomenon, especially parental (low) care and over-protectiveness [14,19,52]. The effects observed for these parental variables on the offspring’s impostor feelings in these studies were of relatively small to moderate size (especially when simultaneously accounting for other socio-emotional variables) and the extent of research evidence was limited. Therefore, while this review work concluded that parent–child relations and parental child-rearing behaviors could play a substantial role in the emergence of the impostor feelings in the offspring, its conclusion was subject to a few considerable limitations and to the acute need for more research evidence. First, it is essential to broaden the limited framework of the familial and parental variables used in previous research to predict impostor feelings, in an attempt to identify patterns that may be more specific to the phenomenon’s context. Based on previous research, it can be assumed that authoritarian parenting could be generally predictive of the child’s impostor feelings, due to its configuration of low care/acceptance and high control. As a specific authoritarian pattern, however, parental psychological control may serve as the mechanism specifically explaining the emergence of impostor feelings (i.e., as a mediator), because of its notorious detrimental nature in affinity to the child’s emotional well-being (as discussed earlier). Indeed, a recent study was the first to demonstrate specifically the unique association between maternal psychological control and students’ impostor feelings [53], both directly and indirectly through low-self-efficacy. Surprisingly, this compelling prediction hypothesis of the impostor phenomenon based on the constellations of these particular parental constructs (i.e., authoritarian parenting and psychological control as mediators) has not yet been tested empirically. Moreover, in light of the acute shortage in studies conducted with adolescents (as the vast majority of studies in this body of research employed adults’ retrospective reports on their family and parents’ patterns), it is also necessary to examine some of the research questions regarding the relationship between parenting factors and the impostor phenomenon with young populations in the present time. Indeed, some highly cited research works have been interested in the long-term effects of past family and parental patterns on the impostor feelings at the present time of adult students or professional workers. In this regard, Sonnak and Towell (2001) [14] demonstrated that perceived past parental over-protectiveness and low care during childhood and adolescence in their original families, along with some aspects of mental health, significantly explained undergraduate adult students’ present impostor feelings. Subject to excluding the participants’ self-esteem as a co-predictor, the findings suggested that parental rearing styles, especially parental (low) care, may play an essential role in the emergence of offspring’s impostor feelings, which persists into adulthood. This implies the existence of long-term consequences of parenting on offspring’s impostor feelings. However, recollections of such patterns are at great risk of inaccuracy and must not be interpreted in longitudinal terms but rather in contemporaneous terms (that is, to treat these retrospective reports as current perceived parenting) [19,54]. Using adolescent participants to test some relative empirical issues could partially overcome this methodological obstacle. Not only could testing etiological-based questions regarding the impostor phenomenon with young populations help illuminate the possible effects of some familial and psycho-social factors during adolescence [11,55], but it could also lay the foundation for testing the longitudinal long-term effects. To date, there have been very few studies in this body of research addressing these crucial needs.

### 1.4. The Current Study

In light of the shortcomings in the etiological research literature about the impostor phenomenon, the current study aimed to test the association between overall maternal and paternal parenting styles (i.e., authoritative, authoritarian, and permissive) and psychological control and impostor feelings among adolescent boys and girls. Surprisingly, these specific parental constructs have not been used in previous research in the empirical context of the impostor phenomenon, despite their terminological centrality and great importance in the body of knowledge of child socialization and parent–child relations in the family. Specifically, the study sought to examine the unique role played by the authoritarian parenting style and psychological control (i.e., when permissive and authoritative parenting are taken into account) in predicting impostor feelings among adolescents. (1) We hypothesized that authoritarian parenting would significantly predict impostor feelings among adolescents, in a form of positive associations between these variables. (2) In addition, since psychological control constitutes a distinct authoritarian parenting pattern that may by particularly detrimental to the child’s emotional well-being, we hypothesized that it would partially mediate the positive association between authoritarian parenting and impostor feelings. Moreover, given the possible differentiation in maternal and paternal parenting styles and practices, as well as in their unique effects on the child’s well-being, the study’s hypotheses were tested separately for fathers and mothers while accounting for the child’s gender. Specifically, the direct and indirect associations (as mediated by parental psychological control) between maternal and paternal authoritarian parenting and adolescents’ impostor feelings were tested for moderation of the child’s gender (in accordance with the model described in Figure 1). The gender of adolescents may have an impact on how they perceive parental authoritarian parenting and psychological control specifically [56]. (3) Therefore, it was expected that the direct and indirect (via parental psychological control) associations between authoritarian parenting and impostor feelings would at least partially vary across adolescent boys and girls (i.e., a moderation and moderated mediation effects by the child’s gender). Yet, the literature on parenting styles provides inconsistent and inconclusive evidence regarding the impact of parenting in parent–adolescent gender dyads [44]. Hence, gender-specific hypotheses (i.e., specifying parent–child gender dyads) on the associations between parenting and impostor feelings cannot be established.

## 2. Methods

### 2.1. Participants and Procedure

Three hundred and eight adolescents took part in an online survey where they reported anonymously on themselves and their parents via several valid psychological questionnaires. The sample consisted of 143 boys and 165 girls whose age ranged from 12 to 17 (*M* = 14.67, SD = 1.64). Age was distributed equally across the gender groups (*F* = 0.12, *p* = 1.64), while boys and girls did not differ by age (*Mean differences* = 0.31; *t*(306) = 1.642, *p* = 0.10). Based on a convenience sampling method, the participants (minor adolescents) were recruited by a professional Israeli survey provider through their parents, who read the research information, perused the questionnaires, and gave their signed consent for their children to take part in an online survey. The participants who received the online link to the survey, subject to their parents’ permission, were introduced to the research details and were presented with an informed consent form, which they were asked to sign prior to filling out the questionnaires. The research procedure and data collection according to this framework were approved beforehand by the author’s institutional review board (IRB) of Tel-Hai academic college (ref. 10-10/2022).

### 2.2. Instruments

Parental Authority Questionnaire (PAQ). The PAQ [57] contains 30 items and is used to classify parents into one of Baumrind’s three parenting styles (Baumrind, 1971), based on the child’s report on a 5-point Likert scale (ranging from 1—strongly disagree to 5—strongly agree): Authoritative (10 items, e.g., “As I was growing up, once family policy had been established, my parents discussed the reasoning behind the policy with the children in the family”), Authoritarian (10 items, e.g., “As I was growing up my parents did not allow me to question any decision they had made”), and Permissive (10 items, e.g., “As I was growing up my parents seldom gave me expectations and guidelines for my behavior”). The index for each of the three parenting styles is the sum of the items of each scale. Thus, the total score for each scale ranges from 10 to 50, with a higher score indicating a higher specification of the parenting style. The questionnaire is widely used internationally for various research purposes. It is a valid questionnaire to assess Baumrind’s (1971) [23] three styles of parenting using adolescents’ reports, with adequate evidence of internal consistency and test–retest reliabilities (0.74 to 0.78) [57,58]. In the current study, we recorded Cronbach’s Alpha coefficients for the permissive, authoritarian, and authoritative scales of 0.77, 0.83, and 0.80 (respectively for father), and 0.77, 0.84, and 0.75 (respectively for mother), consistent with the reliability data reported for the instrument in past research. The scales’ scores appear in Table 1, separately for mothers and fathers.

Impostor Phenomenon Scale (CIPS; Clance, 1985) [5]. The scale contains 20 items designed for self-report, in which the response for an item is given on a 5-point Likert scale. The scale gauges impostor feelings and cognitions, such as fear of evaluation (e.g., “I avoid evaluations if possible and have a dread of others evaluating me”), self-doubt regarding one’s abilities (e.g., “I rarely do a project or task as well as I’d like to do it”), and expressions of phoniness with fears of being exposed by others as a fraudster (e.g., “I’m afraid people important to me may find out that I’m not as capable as they think I am”). The impostor phenomenon construct was originally divided into three theoretical dimensions [5], including self-doubts about one’s own intelligence and abilities (Fake), a tendency to attribute success to chance/luck (Luck), and the inability to admit a good performance (Discount). However, due to the limited support for the three-factor model and the lack of a clearly identifiable factorial structure [59], currently the scoring methodology is commonly used as a unidimensional construct. The Hebrew version of the CIPS (HCIPS) has been validated against external variables, while demonstrating its psychometric properties [60]. Consistent with this evidence, the Cronbach’s Alpha recorded in the current study for the overall scale was 0.92. The scale’s scores appear in Table 1.

Psychological Control Scale–Disrespect (PCDS; Barber et al., 2012) [38]. The parental psychological control of the participants’ mothers and fathers was measured using Barber’s new Psychological Control Scale–Disrespect. Participants were instructed to think about the relationship with their parents during their childhood and adolescence in the family and to determine the extent to which each of the scale’s eight statements (e.g., “my parents try to make me feel guilty for something I’ve done or something they thought I should do”) describe them well (separately for mother and father). The scale was validated against several measures of parenting and child’s outcome, including the child’s antisocial behavior and depression [38]. In the current study, the response for an item was given on a 5-point Likert-type scale ranging from 1—Not like her/him at all to 5—Very much like her/him, with a higher response representing a higher expression of maternal/paternal psychological control. Considering the scale’s relatively small items number, in the current study we obtained good indexes of internal consistency reliability both for the mother (α = 0.82) and father (α = 0.81). The scale’s scores appear in Table 1. The translation and adaptation process of the English PCDS into Hebrew was carried out by the author and a professional bi-lingual English translator, using the three steps back–forward translation procedure.

## 3. Data Analysis

Missing data were handled by employing the listwise deletion method. IBM SPSS statistical package version 28 was used to perform the descriptive and correlational statistics for the sample’s data. For the mediation, moderation, and moderated mediation regression analyses, SPSS macro PROCESS [61] was utilized while applying the bootstrapping method based on the recommendations provided by Preacher and Hayes (2008) [62].

## 4. Results

### 4.1. Preliminary Correlational Analysis and Descriptive Statistics

Table 1 displays the means, standard deviations, and the correlations between the study’s main variables. As expected, adolescents’ impostor feelings were significantly and positively correlated with maternal and paternal authoritarian parenting and with the pattern of psychological control. Impostor feelings were also positively associated with maternal and paternal permissive parenting, but not with authoritative parenting. Parenting styles were intercorrelated significantly, with the authoritative and authoritarian styles inversely associated for mothers and the authoritative and permissive styles positively associated for both parents. Finally, the maternal and paternal parenting styles were correspondingly associated with psychological control, with the latter pattern inversely related to the authoritative style and positively related to the authoritarian style.

### 4.2. Impostor Feelings: Sample Scores and Gender Differences

The sample’s mean score of impostor feelings was below 60, which represents a moderate impostor feelings level [5,63]. Girls reported significantly higher impostor feelings than boys did (*t*(306) = 2.76, *p* < 0.001), with the former scoring on average 56.79 ± 14.83 and the latter scoring on average 51.80 ± 16.85. Further, about 18.8% of the sample’s adolescents scored below 40 (few impostor characteristics), 45.6% scored between 40 and 60 (moderate impostor characteristics), 29.6% scored between 61 and 80 (frequent impostor characteristics), and the rest (about 6%) scored higher than 80 (intense impostor characteristics).

### 4.3. Predicting Impostor Feelings from Parenting Styles and Psychological Control (H1,2)

First, we tested the predictability of the maternal and paternal variables of the impostor feelings in adolescents using a hierarchical multiple regression analysis. At first, the three parenting styles were entered into the regression model as one block, and then the psychological control variable was entered in a subsequent step. This allowed us to weigh the unique contribution added by the latter variable in predicting the adolescents’ impostor feelings from the parental styles and to establish and test its mediating effect in this context. Table 2 presents the results of the regression analysis predicting impostor feelings from the parenting styles (Equation (1)) and psychological control (Equation (2)) for mothers and fathers separately. Consistent with hypothesis 1, the maternal and paternal parenting styles explained a significant proportion of the impostor feelings variance (12.1% and 10.4% respectively), with the authoritarian and permissive parenting styles uniquely and positively correlated with the latter variable. Maternal psychological control predicted an extra significant proportion of 3% of the variance in the child’s impostor feelings (*F* (1, 303) = 11.06, *p* < 0.001), above and beyond the variance explained by the maternal parenting styles. Paternal psychological control predicted an extra significant proportion of 2.9% of the variance in the child’s impostor feelings (*F* (1, 303) = 10.04, *p* = 0.002), above and beyond the variance explained by the paternal parenting styles. Taken together, the maternal and paternal parenting variables explained 15.2% and 13.3% (respectively) of the variance in the adolescents’ impostor scores.

Following the regression analysis, we tested the indirect effect of authoritarian parenting on the impostor feelings via psychological control as a mediator, using a bootstrapping method of a 95% confidence interval. Accordingly, maternal psychological control partially mediated the positive association between maternal authoritarian parenting and adolescents’ impostor feelings, as, after controlling for psychological control, the regression coefficient of authoritarian parenting decreased but was still significant (Equations (1) and (2), Table 2). The positive CI estimate values confirmed the significance of the indirect effect via psychological control as a mediator (*b* = 0.14; CI = 0.04, 0.27). The positive association between paternal authoritarian parenting and adolescents’ impostor feelings was fully mediated by paternal psychological control, as, after controlling for the latter variable, the regression coefficient of the authoritarian parenting decreased and became insignificant (Equations (1) and (2), Table 2). The positive CI estimate values confirmed that the indirect effect via psychological control as a mediator was significant (*b* = 0.15; CI = 0.05, 0.25). Partially consistent with hypothesis 2, the findings indicated that adolescents whose parents are more authoritarian experience greater impostor feelings partially (regarding mothers) and fully (regarding fathers) due to their parents being more psychologically controlling.


*The relationship between authoritarian parenting and adolescent’s impostor feelings: Testing the moderating role of the child’s gender and the moderated mediating role of parental psychological control*

*(H3)*


In this section, we examined whether the main effect of authoritarian parenting and the mediating effect of parental psychological control on the adolescents’ impostor feelings were moderated by the child’s gender (see Figure 1). According to the procedure suggested by Hayes (2013) [61], we used the SPSS macro-PROCESS to test the moderation and moderated mediation for authoritarian parenting on the child’s impostor feelings. The results (Table 3) revealed that the direct positive association between maternal authoritarian parenting and adolescents’ impostor feelings was moderated by the child’s gender (*b* = −0.62, *p* = 0.012), indicating that maternal authoritarian parenting was significantly associated with adolescents’ impostor feelings only among boys (*b* = 0.64, *p* ≤ 0.001) but not among girls (*b* = 0.02, *p* = 0.91). The direct association between paternal authoritarian parenting and adolescents’ impostor feelings was not moderated by the child’s gender (*b* = −0.31, *p* = 0.20), as this association was insignificant among both boys (*b* = 0.36, *p* = 0.06) and girls (*b* = 0.04, *p* = 0.80).

However, the indirect association between maternal authoritarian parenting and impostor feelings was not significantly moderated by the child’s gender (*b* = −0.56, *p* = 0.84), which means that maternal psychological control played a similar mediating role in this association for boys and girls. This was also the case for father–child relations (i.e., the absence of a significant moderated mediation effect for psychological control by the child’s gender; *b* = −0.92, *p* = 0.74), where the association between paternal authoritarian parenting and adolescents’ impostor feelings was fully mediated by psychological control (namely, the paternal effect of authoritarian parenting on impostor feelings, which was only significant through psychological control as mediator, applied equally to boys and girls). Hence, the findings partially support hypothesis 3 embodied in the model (Figure 1) solely for mothers, with the direct effect, but not the indirect effect (as mediated by psychological control), of authoritarian parenting on adolescents’ impostor feelings being moderated by the child’s gender.

## 5. Discussion

Despite the massive growing interest in the impostor phenomenon during the last decade [7], surprisingly little research work has investigated the phenomenon with adolescents [9], and most of it not recently. This is especially peculiar given the knowledge about the phenomenon’s familial roots [1,2] and prior evidence suggesting that impostor feelings in adolescence may be as prevalent and intense as in adulthood [11,64]. Indeed, the current data about adolescent boys and girls recorded similar rates of impostor feelings in comparison with data reported in previous research with adult students [60], with about 36% of the participants reporting experiencing frequent to intense impostor feelings. The present study aimed at reducing this gap in the body of research by testing the association between maternal and paternal authoritarian parenting and impostor feelings among adolescents, while accounting for the mediation effect of parental psychological control and the moderation effect of the child’s gender in this context.

In general, the findings demonstrated the importance of parenting styles and behaviors in concurrently explaining impostor feelings in adolescents, with the permissive and the authoritarian parenting styles uniquely predicting a significant proportion of variance of the adolescents’ impostor scores. Our findings generally accord with previous research data linking between non-authoritative parenting styles (i.e., permissive and authoritarian) and emotional difficulties in adolescents [35]. In the specific emotional context of impostor feelings, these findings could be attributed to the parental properties of care and control [19]. This evidence, however, is the first and the only evidence in the research literature linking between overall parenting styles, according to Baumrind’s typology [23,24] and the impostor phenomenon. The study’s main finding demonstrated the mediating role played by parental psychological control in the association between authoritarian parenting and the impostor phenomenon. While previous research addressed the role played by overcontrolling parenting, including lack of parental care, in the context of the impostor phenomenon [14,19,52], the current findings suggest a more specific explanation regarding the possible mechanism describing the early emergence of impostor feelings in adolescents. In this regard, our data suggest that adolescents whose parents are more authoritarian experience greater impostor feelings due to their parents being more psychologically controlling. Indeed, parental psychological control is described in the parenting literature as a uniquely hazardous pattern to the child’s emotional well-being, normally asserted by the authoritarian parent, whose presence in child-rearing is shown to be associated with several internalized and externalized behavior problems [37,38,40,65] mostly anxiety, depression, and a poor sense of the self. Perhaps the exposure to the prolonged disrespectful, rejecting, conditional, and criticizing nature of psychological control treatment from the parent undermines the child’s self-confidence and self-worth, which, in turn, evokes the dependency on external approval for maintaining a good self-worth. The continuous necessity to conform with the psychologically controlling parent’s expectation to gain approval for the self as worthy in early relations within the family [39,66] could be projected and generalized later in life onto various social and relationship contexts, as the impostor relies on others’ approval and admiration to validate their self-esteem and feelings of self-worth [2].

Interestingly, psychological control as an explaining mechanism of the relationship between authoritarian parenting and the child’s impostor feelings was telling the whole story for fathers but only part of the story for mothers, whose authoritarian parenting effects on impostor feelings was merely partially mediated by psychological control. In other words, relative to mothers, other aspects of authoritarian parenting beyond psychological control per se were relevant to explaining adolescents’ impostor feelings. Along with our finding of a greater proportion of variance of the participants’ impostor feelings explained by the maternal variables compared with the paternal variables (15.2% and 13.3%, respectively), this evidence reinforces the assumption of the gender differential parental effect on the child’s outcome. Specifically, it somewhat reflects a priority of the maternal effects over the paternal effects in predicting impostor feelings (especially concerning authoritarian parenting), as found in some previous research in several contexts of internalizing and externalizing outcomes in adolescents [44,45,46,47,49], but not yet specifically in the context of impostor feelings [10]. Perhaps the fact that mothers are primary caregivers who still play more central and involved roles in child-rearing within the family [10,67] makes the negative influence of maladaptive maternal parenting (especially over psychological control and lack of care) on the child more significant. In the absence of a conclusive picture regarding the impact of parenting in a parent–adolescent dyad by gender in the research literature [44], this generic explanation and more specific ones will need to be further empirically inspected. Based on a hypothesized moderated and moderated mediation model (Figure 1), we also tested the direct and indirect (via psychological control) associations between parental styles and adolescents’ impostor feelings for moderation of the child’s gender. In this regard, a child’s gender moderation was found solely for the maternal direct effect of authoritarian parenting on impostor feelings (i.e., the association between the variables was significant only for boys), but not for the indirect effect through psychological control. Boys may be at greater risk of being more exposed to parental maladaptive behaviors such as severe punishment [68], which can partially explain why only boys’ impostor feelings were affected by mothers’ authoritarian parenting in the current sample. Another explanation could be that boys perceive a higher level of parental psychological control than girls [65]. Importantly, psychological control seemed to have a similar effect across both genders and gender dyads, suggesting that this parental behavior plays a unique, distinct role in the context of adolescents’ impostor feelings, regardless of the child’s gender. Indeed, many researchers have maintained that psychologically controlling parenting has a negative effect on the child’s internal powers such as their sense of self [40,66].

Finally, inconsistent with the data reported in two previous studies with adolescents [11,69], we found gender differences in impostor feelings rates among the sample’s participants, in which girls scored significantly higher than boys. This inconsistency in gender differences across studies could be explained, however, by the lack of uniformity in the research tools used for screening the impostor phenomenon [7]. However, our data are in line with at least 16 previous studies that recorded a gender difference in impostor feelings (mostly with emerging adults) [8], suggesting that, in certain conditions, women rather than men may be more vulnerable to impostor feelings during their lifetime. The question of whether gender differences in impostor feelings are stronger at a younger developmental period (i.e., in adolescence) should be further examined in more studies with young populations, most importantly based on longitudinal data.

The findings of this cross-sectional design study are limited in several respects. First and foremost, these findings and suggested conclusions are not to be interpreted in causal terms, as if parenting styles and behaviors necessarily influence the child’s impostor feelings. Indeed, perhaps the best theoretical way of understanding the current findings regarding the associations between the parenting variables and adolescents’ impostor feelings is in terms of the parental effect on the child. Yet, the study’s data can also suggest the reverse explanation, according to which impostor feelings affect adolescents’ perceptions of their parents’ authoritarian parenting and psychological control. To resolve this possible confounding situation, more longitudinal data must be obtained as part of future research. In that case, it would also be useful to examine the influence of other parenting styles on adolescents’ impostor feelings, which were not measured or considered in the current study. Moreover, the parenting indexes in the current study rely solely on the child’s reports, which reflect their subjective perception and may not match the parent’s self-perceived parenting style nor reflect their actual parenting style [70]. In addition, with the adolescents filling out all the measures, a correlational inflation may potentially be caused by the shared methods measurement [71]. While this could confine the validity of the study findings, there is great merit in the child’s point of view on parental characteristics, due to its importance and close relevance to their behavior and emotion [39,70]. Finally, the study was conducted with Israeli adolescents, hence the generalizability of its findings is limited to the culture and ethnical characteristics in which the current study took place.

## Figures and Tables

**Figure 1 children-10-00308-f001:**
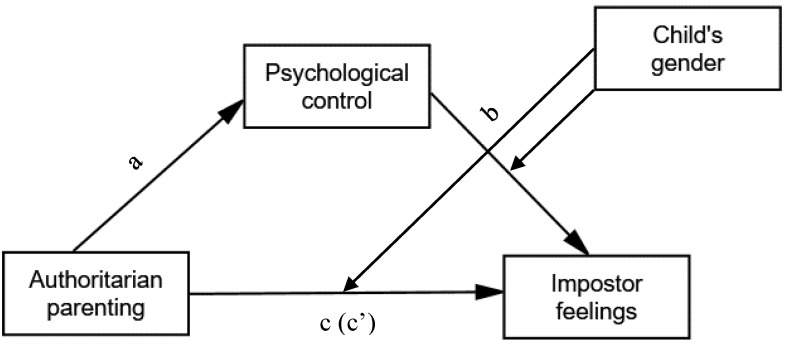
Hypothesized moderation and moderated mediation model for adolescents’ impostor feelings. Direct effect of authoritarian parenting on psychological control (a); direct effect of parental psychological control on adolescents’ impostor feelings (b); total effect of authoritarian parenting on adolescents’ impostor feelings (c); direct effect of authoritarian parenting on adolescents’ impostor feelings, controlling for parental psychological control (c′).

**Table 1 children-10-00308-t001:** Descriptive statistics and zero-order correlations between the study variables (*N* = 308).

	1	2	3	4	5	Mean (Mother)	SD (Mother)
1. Impostor feelings	-	0.28 ***	0.03	0.21 ***	0.24 ***	54.47	15.97
2. Permissive parenting	0.29 ***	-	0.15 **	0.02	0.07	25.76	5.96
3. Authoritative parenting	0.01	0.20 ***	-	−0.18 ***	−0.31 ***	38.18	4.18
4. Authoritarian parenting	0.16 **	0.10	−0.07	-	0.36 ***	29.30	7.05
5. Psychological control	0.26 ***	0.19 ***	−0.32 ***	0.38 ***	-	1.80	0.67
Mean (father)	54.47	25.84	36.98	29.98	1.88	-	
SD (father)	15.97	6.23	5.80	7.24	0.70		-

Note: Figures above diagonal represent mothers’ data and figures below diagonal represent fathers’ data. ** *p* ≤ 0.005, *** *p* ≤ 0.001.

**Table 2 children-10-00308-t002:** Regression analysis predicting the child’s impostor feelings from the parental variables.

	Maternal				Paternal			
	B	SE	t	*p*	B	SE	t	*p*
*Direct effect—Equation (1)*								
Permissive style	0.73	0.15	5.01	<0.001	0.73	0.14	5.12	<0.001
Authoritative style	0.09	0.18	0.48	0.63	−0.11	0.15	−0.73	0.47
Authoritarian style	0.48	0.12	3.84	<0.001	0.30	0.12	2.45	0.015
R^2^		0.121				0.104		
F	*F* (3, 304) = 13.97			*p* < 0.001	*F* (3, 304) = 11.81			*p* < 0.001
*Direct effect—Equation (2)*								
Permissive style	0.68	0.15	4.67	<0.001	0.62	0.15	4.28	<0.001
Authoritative style	−0.25	0.18	−1.39	0.16	0.08	0.16	0.48	0.63
Authoritarian style	0.34	0.13	2.62	0.009	0.150	0.14	1.18	0.24
Psychological control	4.69	1.41	3.33	<0.001	4.56	1.44	3.17	0.002
R^2^	0.152 *F* (4, 303) = 13.59					0.133		
F				*p* < 0.001	*F* (4, 303) = 11.63			*p* < 0.001

**Table 3 children-10-00308-t003:** Moderation and moderated mediation analysis for impostor feelings: Regression model predicting impostor feelings from authoritarian parenting and psychological control as a mediator with child’s gender as a moderator.

	Maternal				Paternal			
	B	SE	t	*p*	B	SE	t	*p*
Authoritarian style	1.26	0.40	3.17	0.002	0.67	0.40	1.68	0.09
Parental-psychological control	4.66	1.28	3.38	<0.001	0.53	1.35	3.90	<0.001
Child’s gender	23.25	7.36	3.16	0.002	14.76	7.45	1.98	0.049
Authoritarian style X Child’s gender	−0.62	0.24	−2.53	0.012	−0.31	0.24	−1.29	0.20
Psychological control X Child’s gender	−0.56	2.76	−0.20	0.84	−0.92	2.72	−0.34	0.74

## Data Availability

Data is available from the author upon reasonable request.

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
