# Peer review of "Maternal and Paternal Authoritarian Parenting and Adolescents’ Impostor Feelings: The Mediating Role of Parental Psychological Control and the Moderating Role of Child’s Gender"

_children, 2023, doi:10.3390/children10020308_

Round 1
Reviewer 1 Report
Manuscript Number 2175390 Title: “Maternal and Paternal Authoritarian Parenting and Adolescents’ Impostor Feelings: The Mediating Role of Parental Psychological Control and the Moderating Role of Child’s Gender”
The present study investigates the association between maternal and paternal authoritarian parenting and impostor feelings among adolescents, while testing the mediating role played by pa-rental psychological control and the moderating role of the child’s gender in this context. Sample was composed of 308 adolescents (143 boys and 165 girls) from Israel. Results showed that parenting styles are associated with the adolescents’ impostor feelings and parental psychological control mediated the relation between parental authoritarian parenting and the adolescents’ impostor feelings.
The manuscript addresses an interesting and widely studied topic in the literature, i.e., the relationship between parental characteristics and child adjustment. However, the theoretical conceptualization is rather weak and needs to be improved. In addition, some empirical corrections should be made.
Theoretical part
The present manuscript examines the relationship between parental characteristics during the socialization process (parenting styes) and psychosocial adjustment of the child (measured by impostor feelings). There is much literature, classic and recent, about the relationship between parental characteristics and psychosocial adjustment of the child. Authors should add a further theoretical conceptualization in the introduction section including the two-dimensional model (Maccoby and Martin, 1983) (including parenting dimensions and parenting styles) and classical and recent evidence about parental socialization and its relation to child adjustment.
In the manuscript, the two-dimensional orthogonal model is mentioned without reference to its authors, Maccoby and Martin. Moreover, it is presented in a very general and confusing way that could imply that its author is Baumrind and not Maccoby and Martin. The conceptualization of parental styles proposed in the manuscript based on two parental dimensions (responsiveness and demandingness) is not typical of Baumrind's model, but of Maccoby's model. The authors should clarify this in the introduction section. The authors should clearly distinguish Baumrind's Y-model and Maccoby and Martin's two-dimensional theoretical model.
Parental socialization refers to the process by which the adult (i.e., parents) can transmit to the young person (i.e., child) the habits and values of the culture of origin so that the child adopts adequate functioning within the culture to which the child belongs (Climent-Galarza et al., 2022, Sandoval-Obando et al., 2022, Martinez-Escudero et al., 2020, Maccoby and Martin, 1983). There are many models and studies on parental socialization. One of the earliest models was Baumrind´s Y model (Baumrind, 1968). Baumrind's Y model (Baumrind, 1968) proposed three parental styles: authoritative, authoritarian, and permissive; which corresponded to three modes of parental control, the authoritative control, the authoritarian control, and the lack of control (i.e., permissive control) (Baumrind, 1968).
Subsequently, Maccoby and Martin's two-dimensional model (Maccoby and Martin, 1983) emerged and became the reference model in the parental socialization literature and has given rise to much empirical evidence on parental styles and child adjustment. Maccoby and Martin's two-dimensional model state that parents use two independent dimensions to socialize their children and to relate with them (i.e., warmth and strictness) (Climent-Galarza et al., 2022, Fuentes et al., 2022, Gimenez-Serrano et al., 2022, Palacios et al., 2022, Martínez et al., 2021, Martinez et al., 2020, Queiroz et al., 2020, Garcia and Gracia, 2009, Darling and Steinberg, 1993, Lamborn et al., 1991, Maccoby and Martin, 1983). Parental warmth is also labeled responsiveness (as in the present study), involvement, acceptance or implication (Martinez et al., 2020, Garcia and Gracia, 2014, Darling and Steinberg, 1993), or affection (Martinez-Escudero et al., 2020) and refers to the extent of parents show the children love, approval, acceptance and affection, give them their support (Gimenez-Serrano et al., 2022), sensitivity, and affection when dealing with their children, use dialogue with their children (Climent-Galarza et al., 2022), communication and reasoning with them (Martinez et al., 2020, Martínez et al., 2019). Parental strictness is also labeled demandingness (as in the present study), control, firmness (Steinberg, 2005, Darling and Steinberg, 1993), imposition (Martinez-Escudero et al., 2020) or supervision (Garcia et al., 2020) and refers to the extent of parents use discipline towards their children, controlling and/or supervising their behavior (Gimenez-Serrano et al., 2022), establishing norms for children’s behavior, and maintaining position of authority (Darling and Steinberg, 1993, Baumrind, 1991b) and parental demands placed on children to promote compliance, i.e., the degree of imposition, authority, or rigidity (Climent-Galarza et al., 2022). According to Maccoby and Martin's model, four parenting styles emerges from the combination of the two main parenting dimensions (i.e., warmth and strictness): authoritarian (strictness but not warmth); authoritative (strictness and warmth), indulgent (warmth but not strictness) and neglectful (neither strictness nor warmth) (Climent-Galarza et al., 2022, Fuentes et al., 2022, Gimenez-Serrano et al., 2022, Palacios et al., 2022, Garcia et al., 2020, Perez-Gramaje et al., 2020, Queiroz et al., 2020, Villarejo et al., 2020, Darling and Steinberg, 1993, Maccoby and Martin, 1983).
Many studies have focused on investigating which parenting style is best and this has resulted in an extensive literature on this topic.
Classical studies conducted in Anglo-Saxon contexts with European-American samples (mostly white middle-class families) state that the authoritative style (i.e., combination of parental warmth and parental strictness together) is related to the best child psychosocial adjustment (Steinberg et al., 1994, Darling and Steinberg, 1993, Baumrind, 1991a, Lamborn et al., 1991, Steinberg et al., 1991). This is not always true in all cultural contexts. Other studies conducted in ethnic minority groups in the United States such as Chinese Americans (Chao, 2001) or African American (Deater-Deckard et al., 1996), and Arabs societies (Dwairy and Achoui, 2006) state that authoritarian style (i.e., parental strictness without parental warmth) is related to the best child adjustment.
Contrary to classical studies, the most recent evidence conducted in European and Latin American countries support the idea that indulgent style (i.e., parental warmth without parental strictness, similar to the labeled permissive style in the present study) is associated with the best child psychosocial adjustment (Fuentes et al., 2022, Gimenez-Serrano et al., 2022, Palacios et al., 2022, Gimenez-Serrano et al., 2021, Garcia et al., 2020, Martinez-Escudero et al., 2020, Perez-Gramaje et al., 2020, Villarejo et al., 2020, Garcia and Serra, 2019, Martínez et al., 2019, Garcia et al., 2018, Garcia and Gracia, 2009).
According to the manuscript, “authoritative and permissive (high on responsiveness) parents vary on demandingness variables” and “Authoritative parents (high on both dimensions)”. So, permissive style is high on responsiveness but low in demandingness (like indulgent parenting in the bidimensional model of Maccoby and Martin. Authors should clarify it to understand better the evaluated variables.
Empirical part
According to most scientific articles, the organization of section 2. Method includes three subsections: sample and procedure, measures or instruments and plan of analysis. However, the manuscript only has *sample and procedure* and *measures or instruments*. Authors should add a subsection called *plan of analysis* within section 2. Method. The * plan of analysis* section comprises the statistical analyses performed on the data.
Baumrind, D. (1991a). "Effective parenting during the early adolescent transition," Advances in Family Research Series. Family Transitions. eds. P.A. Cowan, and E.M. Herington (Hillsdale, NJ, US: Lawrence Erlbaum Associates, Inc), 111-163.
Baumrind, D. (1991b). "Parenting styles and adolescent development," Encyclopedia of Adolescence. eds. R.M. Lerner, A.C. Petersen, J. Brooks-Gunn (New York: Garland), 746-758.
Baumrind, D. (1968). Authoritarian vs authoritative parental control. Adolescence 3, 255-272.
Chao, R.K. (2001). Extending research on the consequences of parenting style for Chinese Americans and European Americans. Child Dev. 72, 1832-1843. doi: 10.1111/1467-8624.00381.
Climent-Galarza, S., Alcaide, M., Garcia, O.F., Chen, F., Garcia, F. (2022). Parental socialization, delinquency during adolescence and adjustment in adolescents and adult children. Behavioral Sciences 12. doi: 10.3390/bs12110448.
Darling, N., and Steinberg, L. (1993). Parenting style as context: An integrative model. Psychol. Bull. 113, 487-496. doi: 10.1037/0033-2909.113.3.487.
Deater-Deckard, K., Dodge, K.A., Bates, J.E., Pettit, G.S. (1996). Physical discipline among African American and European American mothers: Links to children's externalizing behaviors. Dev. Psychol. 32, 1065-1072. doi: 10.1037/0012-1649.32.6.1065.
Dwairy, M., and Achoui, M. (2006). Introduction to three cross-regional research studies on parenting styles, individuation, and mental health in Arab societies. J. Cross-Cult. Psychol. 37, 221-229. doi: 10.1177/0022022106286921.
Fuentes, M.C., Garcia, O.F., Alcaide, M., Garcia-Ros, R., Garcia, F. (2022). Analyzing when parental warmth but without parental strictness leads to more adolescent empathy and self-concept: Evidence from Spanish homes. Front. Psychol. 13. doi: 10.3389/fpsyg.2022.1060821.
Garcia, F., and Gracia, E. (2014). "The indulgent parenting style and developmental outcomes in South European and Latin American countries," Parenting Across Cultures. ed. H. Selin (Dordrecht, Netherlands: Springer), 419-433.
Garcia, F., and Gracia, E. (2009). Is always authoritative the optimum parenting style? Evidence from Spanish families. Adolescence 44(173), 101-131.
Garcia, O.F., Fuentes, M.C., Gracia, E., Serra, E., Garcia, F. (2020). Parenting warmth and strictness across three generations: Parenting styles and psychosocial adjustment. Int. J. Environ. Res. Public Health 17, 7487. doi: 10.3390/ijerph17207487.
Garcia, O.F., and Serra, E. (2019). Raising children with poor school performance: Parenting styles and short- and long-term consequences for adolescent and adult development. Int. J. Environ. Res. Public Health 16, 1089. doi: 10.3390/ijerph16071089.
Garcia, O.F., Serra, E., Zacares, J.J., Garcia, F. (2018). Parenting styles and short- and long-term socialization outcomes: A study among Spanish adolescents and older adults. Psychosoc. Interv. 27, 153-161. doi: 10.5093/pi2018a21.
Gimenez-Serrano, S., Alcaide, M., Reyes, M., Zacarés, J.J., Celdrán, M. (2022). Beyond parenting socialization years: The relationship between parenting dimensions and grandparenting functioning. Int. J. Environ. Res. Public Health 19, 4528. doi: 10.3390/ijerph19084528.
Gimenez-Serrano, S., Garcia, F., Garcia, O.F. (2021). Parenting styles and its relations with personal and social adjustment beyond adolescence: Is the current evidence enough? Eur. J. Dev. Psychol. 19, 749-769. doi: 10.1080/17405629.2021.1952863.
Lamborn, S.D., Mounts, N.S., Steinberg, L., Dornbusch, S.M. (1991). Patterns of competence and adjustment among adolescents from authoritative, authoritarian, indulgent, and neglectful families. Child Dev. 62, 1049-1065. doi: 10.1111/j.1467-8624.1991.tb01588.x.
Maccoby, E.E., and Martin, J.A. (1983). "Socialization in the context of the family: Parent–child interaction," Handbook of Child Psychology. ed. P.H. Mussen (New York: Wiley), 1-101.
Martínez, I., Murgui, S., Garcia, O.F., Garcia, F. (2021). Parenting and adolescent adjustment: The mediational role of family self-esteem. J. Child Fam. Stud. 30, 1184-1197. doi: 10.1007/s10826-021-01937-z.
Martínez, I., Murgui, S., Garcia, O.F., Garcia, F. (2019). Parenting in the digital era: Protective and risk parenting styles for traditional bullying and cyberbullying victimization. Comput. Hum. Behav. 90, 84-92. doi: 10.1016/j.chb.2018.08.036.
Martinez, I., Garcia, F., Veiga, F., Garcia, O.F., Rodrigues, Y., Serra, E. (2020). Parenting styles, internalization of values and self-esteem: A cross-cultural study in Spain, Portugal and Brazil. Int. J. Environ. Res. Public Health 17, 2370. doi: 10.3390/ijerph17072370.
Martinez-Escudero, J.A., Villarejo, S., Garcia, O.F., Garcia, F. (2020). Parental socialization and its impact across the lifespan. Behav. Sci. 10, 101. doi: 10.3390/bs10060101.
Palacios, I., Garcia, O.F., Alcaide, M., Garcia, F. (2022). Positive parenting style and positive health beyond the authoritative: Self, universalism values, and protection against emotional vulnerability from Spanish adolescents and adult children. Front. Psychol. 13. doi: 10.3389/fpsyg.2022.1066282.
Perez-Gramaje, A.F., Garcia, O.F., Reyes, M., Serra, E., Garcia, F. (2020). Parenting styles and aggressive adolescents: Relationships with self-esteem and personal maladjustment. Eur. J. Psychol. Appl. Legal Context 12, 1-10. doi: 10.5093/ejpalc2020a1.
Queiroz, P., Garcia, O.F., Garcia, F., Zacares, J.J., Camino, C. (2020). Self and nature: Parental socialization, self-esteem, and environmental values in Spanish adolescents. Int. J. Environ. Res. Public Health 17, 3732. doi: 10.3390/ijerph17103732.
Sandoval-Obando, E., Alcaide, M., Salazar-Muñoz, M., Peña-Troncoso, S., Hernández-Mosqueira, C., Gimenez-Serrano, S. (2022). Raising children in risk neighborhoods from Chile: Examining the relationship between parenting stress and parental adjustment. Int. J. Environ. Res. Public Health 19. doi: 10.3390/ijerph19010045.
Steinberg, L. (2005). "Psychological control: Style or substance?" New Directions for Child and Adolescent Development: Changes in Parental Authority during Adolescence. ed. J.G. Smetana (San Francisco: Jossey-Bass), 71-78.
Steinberg, L., Lamborn, S.D., Darling, N., Mounts, N.S., Dornbusch, S.M. (1994). Over-Time changes in adjustment and competence among adolescents from authoritative, authoritarian, indulgent, and neglectful families. Child Dev. 65, 754-770. doi: 10.1111/j.1467-8624.1994.tb00781.x.
Steinberg, L., Mounts, N.S., Lamborn, S.D., Dornbusch, S.M. (1991). Authoritative parenting and adolescent adjustment across varied ecological niches. J. Res. Adolesc. 1, 19-36.
Villarejo, S., Martinez-Escudero, J.A., Garcia, O.F. (2020). Parenting styles and their contribution to children personal and social adjustment. Ansiedad Estres 26, 1-8. doi: 10.1016/j.anyes.2019.12.001.

Reviewer 2 Report
The authors investigated the association between maternal and paternal authoritarian parenting and impostor feelings among adolescents. Overall, the manuscript is well-written and I only have several minor comments:
1) Introduction: (a) More explanation/elaboration is needed for the mediating role of psychological control in the association between authoritarian parenting and impostor feelings. (b) Why only authoritarian parenting is proposed in the moderated mediation model (see Figure 1), but not permissive and authoritative parenting?
2) The current study: (a) Please provide theoretical support for the moderated mediation model in Figure 1. (b) Kindly mention all the hypotheses so that the readers know specifically what research questions this study is answering.
3) Participants and procedure: What is the sampling method?
4) Methods: Please include a Data Analysis section to describe the statistical analyses to be conducted to verify each hypothesis.
5) Results: After adding all the hypotheses, kindly organize the results accordingly, so that readers could easily know whether each hypothesis is supported.
6) Discussion: Suggest adding practical implications of this study.
Reviewer 3 Report
This is a strong paper with well-done statistical analyses and important conclusions for the field. There are some suggestions for improvement:
1) In the first paragraph of the section on parenting styles, when the responsiveness and demandingness dimensions are explained, the descriptions of the parenting styles should state how both dimensions correspond to each style (i.e., authoritarian is high on demandingness and low on responsiveness, permissive is high on responsiveness and low on demandingness).
2) The section on parenting styles should explain with research support the benefits of authoritative parenting and the harms of permissive parenting since both are looked at in the study.
3) The paragraph on Parenting and the Impostor Phenomenon mentions "highly cited research." This research should be explained.
4) It is unclear in this paragraph how and why retrospective reports can be treated as current if they are retrospective.
5) The authors should cite any previous research on impostor syndrome in adolescents if it is available in the literature review.
6) For the Parental Authority Questionnaire, what are the anchors for the Likert scale?
7) There needs to be an example item for expressions of phoniness for the Impostor Phenomenon Scale.
8) It is unclear why for the Psychological Control Scale the adolescents rated psychological control over their childhood and adolescence. It seemed that a current rating would be more accurate.
9) In the Results section on moderation, direct moderation results were tested and reported for mothers but not fathers. The fathers' results for direct effects with moderation should be reported.
10) In the second paragraph of the Discussion, the reasons why Authoritarian and Permissive parenting explained variance in the impostor syndrome should be explained in more detail with research support.
11) The authors make a strong claim in the Discussion of how mothers' effects are stronger than fathers' effects in predicting the imposter syndrome when the difference in variance explained is less than 2%. The strength of this assertion should be modified based on this small number.
12) The authors claim that psychological control has a similar effect on both genders after explaining moderation effects for mothers. These statements appear to contradict.
13) In the Limitations section, it should be mentioned that there is shared method variance with the adolescents filling out all measures.
14) In the Limitations section, the generalizability limitations to other cultures, ethnicities, and countries should be mentioned.
Round 2
Reviewer 1 Report
Manuscript Number 2175390
Title: “Maternal and Paternal Authoritarian Parenting and Adolescents’ Impostor Feelings: The Mediating Role of Parental Psychological Control and the Moderating Role of Child’s Gender”
The authors have made theoretical and empirical modifications that have improved the quality of the manuscript. Nevertheless, some modifications should be made.
In the introduction, the parenting styles named in the manuscript (i.e., authoritarian, authoritative and permissive) should be described on the basis of their dimensions (i.e., warmth and strictness). For example, permissive parents are warm but not strict.
It is also convenient to add in the theoretical conceptualization (introduction) the benefits of other parental styles such as indulgent and authoritative.
Classical studies conducted in Anglo-Saxon contexts with European-American samples (mostly white middle-class families) state that the authoritative style (i.e., combination of parental warmth and parental strictness together) is related to the best child psychosocial adjustment (Darling and Steinberg, 1993, Lamborn et al., 1991).
Contrary to classical studies, the most recent evidence conducted in European and Latin American countries support the idea that indulgent style (i.e., parental warmth without parental strictness, similar to the labeled permissive style in the present study) is associated with the best child psychosocial adjustment (Fuentes et al., 2022, Palacios et al., 2022).
In the discussion section it should be added as a limitation that other parenting styles high in warmth have not been measured: the style characterized by high warmth and low strictness (Fuentes et al., 2022, Palacios et al., 2022) and the style characterized by high warmth and high strictness (Darling and Steinberg, 1993, Lamborn et al., 1991).
References
Darling, N., and Steinberg, L. (1993). Parenting style as context: An integrative model. Psychol. Bull. 113, 487-496. doi: 10.1037/0033-2909.113.3.487.
Fuentes, M.C., Garcia, O.F., Alcaide, M., Garcia-Ros, R., Garcia, F. (2022). Analyzing when parental warmth but without parental strictness leads to more adolescent empathy and self-concept: Evidence from Spanish homes. Front. Psychol. 13. doi: 10.3389/fpsyg.2022.1060821.
Lamborn, S.D., Mounts, N.S., Steinberg, L., Dornbusch, S.M. (1991). Patterns of competence and adjustment among adolescents from authoritative, authoritarian, indulgent, and neglectful families. Child Dev. 62, 1049-1065. doi: 10.1111/j.1467-8624.1991.tb01588.x.
Palacios, I., Garcia, O.F., Alcaide, M., Garcia, F. (2022). Positive parenting style and positive health beyond the authoritative: Self, universalism values, and protection against emotional vulnerability from Spanish adolescents and adult children. Front. Psychol. 13. doi: 10.3389/fpsyg.2022.1066282.

Author Response
Thank you very much!

Reviewer 3 Report
All reviewer comments have been answered adequately.
Author Response
Thank you very much!